# The Emotion of Disgust among Medical and Psychology Students

**DOI:** 10.3390/diseases8040043

**Published:** 2020-11-15

**Authors:** Artemios Pehlivanidis, Niki Pehlivanidi, Katerina Papanikolaou, Vassileios Mantas, Elpida Bertou, Theodoros Chalimourdas, Vana Sypsa, Charalambos Papageorgiou

**Affiliations:** 11st Department of Psychiatry, Medical School-National and Kapodistrian University of Athens, Eginition Hospital, Vas Sophias 72, 11528 Athens, Greece; mantas.vasilios@yahoo.com (V.M.); thodoris.chalimourdas@gmail.com (T.C.); chpapag@med.uoa.gr (C.P.); 2Medical School-National and Kapodistrian University of Athens, Eginition Hospital, Vas Sophias 72, 11528 Athens, Greece; nikipehlivanidi99@gmail.com; 3Department of Child Psychiatry, Medical School-National and Kapodistrian University of Athens, “Agia Sophia” Children’s Hospital, Thivon and Papadiamantopoulou Street, 11527 Athens, Greece; katpapan@med.uoa.gr; 4School of Psychology-National and Kapodistrian University of Athens, Eginition Hospital, Vas Sophias 72, 11528 Athens, Greece; elpiss.btt@gmail.com; 5Department of Hygiene, Epidemiology and Medical Statistics, Medical School, National and Kapodistrian University of Athens, Mikras Asias 75, 11527 Athens, Greece; vsipsa@med.uoa.gr

**Keywords:** disgust, DS-R, medical students, psychology students, academic orientation, specialization

## Abstract

The emotion of disgust evolved as a way to protect oneself from illness and is associated with aspects of disease avoidance. Disgust Scale–Revised (DS-R) (Olatunji et al., 2008) measures the disgust propensity of three kinds of disgust (core, animal reminder, contamination). Contextual factors, such as academic background, might influence DS-R scoring, especially among medical students, where the notion of disease is central. We examined DS-R scoring and the choice of postgraduate studies in medical (*n* = 94) and psychology (*n* = 97) students. In an anonymous web-based survey, participants completed the DS-R and a questionnaire including plans for postgraduate studies. Females outnumbered males and scored higher in total DS-R score (*p* = 0.003). Psychology students scored higher in all three kinds of disgust (*p* < 0.001 for core disgust and animal reminder, *p* = 0.069 for contamination disgust), indicating a higher level of disease avoidance. Medical students willing to follow Internal Medicine scored higher in core disgust (*p* < 0.05), while psychology students willing to study Experimental Psychology scored lower in the animal reminder subscale (*p* = 0.019 and *p* < 0.001 for the association between these subscales and the orientation of Medical and Psychology Students, respectively). In conclusion, disgust propensity as rated by DS-R is related to academic background and orientation preferences in postgraduate studies.

## 1. Introduction

Disgust is one of the basic emotions in Plutchik’s (1980) [1] theory of emotions and there is strong evidence that it evolved as a way to protect one’s self from illness [2,3]. Disgust has been extensively studied by Rozin et al. (1987) [4], who proposed a classification system of four broad categories of disgust elicitors. Core elicitors are characterized by a real or perceived threat of oral incorporation and a reactive sense of offensiveness (e.g., spoiled milk, feces, and rats). Animal reminder elicitors consist of our own mortality and inherent animalistic nature (attitudes and practices surrounding sex, injury of the body or violations of its outer envelope and death). Interpersonal disgust is elicited by contact with individuals who are unknown, ill or tainted by disease, misfortune or immorality. Socio-moral disgust elicitors is a subclass of moral violations that the person is morally “sick” of or “twisted”. Haidt et al. (1994) [5] developed the Disgust Scale (DS), with eight subscales. The Disgust Scale—Revised (DS-R) provided by Olatunji et al. (2008) [6] revealed three distinct factors with a better internal consistency and conceptualized three subscales, namely core, animal reminder and contamination disgust. The core disgust factor is primarily characterized as a food rejection response centered on oral incorporation of offensive stimuli (e.g., eating monkey meat). The animal reminder factor highlights stimuli or behaviors that serve as reminders of the origins of humans (e.g., touching a dead body). The contamination disgust factor (e.g., accidentally drinking from someone else’s cup) reflects disease spread by people, whereas core disgust reflects disease spread by objects. The DS-R scale has been validated and its three factors were confirmed internationally, including a Korean sample [7], students in the Netherlands [8], samples from the United States [9], Israel [10], Greece [11] and samples from Brazil, Australia, Germany, Italy, Japan, and the Netherlands [12]. 

However, there is still need for further investigation on the influence of demographic characteristics on disgust propensity. Berger and Anaki (2014) [10], in a large heterogeneous sample (*n* = 1427), investigated the role of demographic variables on disgust’s sensitivity. Gender was found to have a large effect on DS-R score, while the effects of other demographic variables, such as religion, political view, education and age, were exceptionally modest. They suggest that demographic variables create some of the diverse contexts in which disgust is evoked, without modulating the intensity of the subjective disgust sensitivity. Chalimourdas et al. (2019) [11] administered the Greek version of the DS-R in a sample of 754 healthy participants. Women scored higher than men in all three subscales and religiousness was associated with heightened levels of disgust. There was a weak negative correlation between levels of disgust and years of education. 

Nevertheless, in previous reports the academic context has not been studied for its impact on the emotion of disgust as measured by the DS-R scale. DS-R was refined mainly with psychology students. Students of a different academic background, and especially medical students, who are expected to be exposed to diseases more than any other students, might score differently in DS-R. The data suggest marginally lower disgust sensitivity among medical students and that medical training may reduce sensitivity [13]. Furthermore, the assessment of students’ future plans for postgraduate studies might reveal differences among the three disgust subscales, since different orientations involve distinct patterns of exposure to disgust-eliciting stimuli. 

Studies suggest that decisions regarding medical specialization have been associated with demographic and outcome-expectancy type variables [14], personality-type characteristics [15], institutional factors and personal values [16], exposure to role models [17]. Reports examining the role of disgust sensitivity in decisions regarding post-graduate career specializations are scarce. According to Consedine and Windsor (2014) [18], disgust sensitivity predicted interest in careers of varying procedural intensity among medical students. 

The present study has a twofold aim. First, to compare a sample of medical students to a sample of psychology students in order to assess the impact of academic background on DS-R scoring. Second, to investigate their future plan for postgraduate studies in relation to DS-R scoring. Our hypotheses were that educational orientation, while implying a different attitude towards disease avoidance, would elicit different disgust scores, and that postgraduate orientation with less human contact and treatment (such as Experimental Psychology or Laboratory Specialty) might elicit different DS-R scores in the two samples of students.

## 2. Materials and Methods

### 2.1. Design and Participants

During a two-week period at the beginning of the academic period 2019–2020, 94 medical students and 97 psychology students of the National and Kapodistrian University of Athens (NKUA) responded to a web-based survey, which guaranteed their anonymity. Participants were reached by one of their classmates through their University Social Media groups. Students are familiar with web-based approaches, and having their classmates as contact persons instead of their professors possibly increases the objectivity of the responses. All students completed the DS-R and a questionnaire comprising their demographics and their plans for postgraduate studies. All subjects gave their informed consent for inclusion before they participated in the study. The study was conducted in accordance with the Declaration of Helsinki, and the protocol was approved by the Ethics Committee of the 1st Department of Psychiatry of the NKUA, Eginition Hospital (ADA: BIPM46Ψ8NZ-OΩΩ). 

### 2.2. Instruments

The anonymous questionnaire comprised information on sex, age and year of study. Medical students had to choose one out of among three potential future options of medical specialty, namely, Internal Medicine, Surgical Specialty, Laboratory Specialty, while psychology students had to choose one of the three following postgraduate study directions: Clinical Psychology, Educational Psychology and Experimental Psychology.

### 2.3. The DS-R Scale

The scale consists of three subscales (a 12-item Core disgust scale, an eight-item Animal reminder scale and a five-item Contamination disgust scale) with good psychometric characteristics [6] The Greek version of DS-R has been examined in a sample consisting of 754 individuals originating from all nine administrative divisions of Greece [11]. The Exploratory Factor Analysis (EFA) method determined a three-factor solution with similar structure to that of the original three-factor model, presented by Olatunji et al. [6]. The two-factor EFA solution also had an adequate fit to our data. All reliability indices of the scale were satisfactory. With respect to the test–retest reliability, the Interclass Correlation Coefficient (ICC) ranged between 0.85 and 0.92 (*p* < 0.001 in all cases) confirming the stability of the scale. The Cronbach’s alpha of the total scale was 0.84 for the total scale reflecting satisfactory internal consistency. 

### 2.4. Statistics

Participants’ characteristics were described using mean and standard deviation for quantitative variables or proportions for categorical variables. DS-R scores were summarized using means and standard deviation. 

We analysed the internal consistency of the items using Cronbach’s α after reversing the score of questions 1, 6 and 10, for which higher scores indicated less disgust. We tested for measurement and structural invariance of the disgust scale based on the three subscales (core disgust, animal reminder and contamination disgust) across the two groups of students participating in the study. We used a forward approach, starting with a model assuming total lack of invariance and adding restrictions for equality of specific parameters across groups (factor loadings, item intercepts, residual variances, factor means and factor variances). We used chi-square tests to compare the nested models. 

The demographic characteristics of the two groups of students as well as their DS-R scores were compared using *t*-test, chi-squared test or Mann–Whitney test, as appropriate. Multiple logistic regression models were used to assess factors independently associated with reporting Laboratory Specialty as future option (among medical students) and post-graduate Experimental orientation (among psychology students) compared to all the other available options.

## 3. Results

Completed data were available for 94 medical and 97 psychology students (total *n* = 191). Cronbach’s α for the 27 items of the scale was 0.86. While testing for measurement and structural invariance of the DS-R, we identified metric invariance (equivalent factor loading), partial strong (intercepts) and partial strict (residuals) invariance of the disgust scale based on the three subscales. Approximately three quarters of the students (77.5%, *n* = 148) were females. Their mean (SD) age was 20.2 (2.2) years. A total of 121 (63.4%) were students in the introductory two years of their studies. Among medical students, Internal Medicine and Surgical Specialty were the most frequently reported future options (42.5% and 47.9%, respectively) while Laboratory Specialty was an option only for 9.6% of them. Among psychology students, 67.0% chose Clinical Psychology as a post-graduate option, 21.7% chose Educational Psychology and 11.3% Experimental Psychology. The two groups did not differ in age (Table 1). A larger number of medical students in comparison to psychology students were males and belonged to an advanced study year. Medical students scored lower in all DS-R subscales compared to psychology students (*p* < 0.001 for core disgust and animal reminder, marginally statistically significant for contamination disgust, *p* = 0.069), with a mean (SD) total score of 49.57 (13.54) and 62.71 (14.57), respectively (*p* < 0.001) (Table 1). Medical students with Internal Medicine orientation scored higher in the core disgust subscale compared to Surgical and Laboratory Specialty (mean (SD): 30.73 (5.30), 2.44 (7.80) and 28.22 (7.87), respectively, *p* = 0.019) (Table 2). Psychology students with the Experimental postgraduate orientation scored lower in the animal reminder subscale (Table 2 and Table 3). The score distribution of DS-R subscales did not differ between the two faculty students in relation to sex (Table 4).

Among medical students, there was no association between age, sex, study year and DS-R subscales with the odds of reporting Laboratory Specialty as future option compared to the other specialties (data not shown). Among psychology students, from a multiple logistic regression model, females and students with higher total disgust score had a lower probability of reporting post-graduate Experimental Psychology orientation as compared to Clinical Psychology or Educational Psychology. Conversely, the students scoring higher in the Core disgust subscale had a higher probability of choosing Experimental orientation (Table 5).

## 4. Discussion

The first aim of the study was to explore possible differences in DS-R scoring between medical and psychology students. Medical students scored lower in all DS-R subscales in comparison to psychology students indicating that educational orientation, might have an impact on DS-R scoring. This finding challenges the notion supported by Berger and Anaki (2014) [10] that, with the exception of gender, the effect of demographic variables on the intensity of the subjective disgust sensitivity is exceptionally modest. Our finding is in accordance with data supporting lower disgust sensitivity among medical students [13]. It seems that medical students are less sensitive to a reactive sense of offensiveness (e.g., spoiled milk, feces), attitudes and practices surrounding injury of the body or violations of its outer envelope, death and contamination. Protection from illness has been considered as one of the evolving roles of disgust [2,3]. Disgust has been found to increase during the first trimester of pregnancy [19] and certain aspects of disgust have been linked to mate preference [20,21]. These findings suggest that disgust is associated with various aspects of reproduction and disease avoidance. Since medical students are expected to be less avoidant of diseases, differences in scoring between medical and psychology students may be attributed to choices when deciding to enter undergraduate studies. The decision to follow psychology studies may be influenced by the basic emotion of disgust as it is measured by DS-R in opposition to that of medical students. It might be that students of other academic backgrounds with less human contact and minimal exposure to physical or mental diseases would score differently, possibly even higher, in the DS-R, reflecting a different attitude towards disease avoidance. A study closer to the decision making for undergraduate studies could help to further elucidate the influence of the emotion of disgust on the decision-making for study orientation. 

In our study, DS-R score distribution was not influenced by sex of either medical or psychology students, while, in the DS-R score distribution of all students in relation to sex, females scored higher. This could be explained by the larger sample when these two groups of students were combined. Female preponderance in disgust sensitivity is a standard finding in previous reports [22,23]. As mentioned above, evolutionarily, women use more energy and resources to give birth and raise their children [22] and have to protect themselves and their children from illness. In addition, Spark et al. (2018) [24], in an extensive meta-analysis documenting the sex difference in disgust, argue that key features of this pattern are best explained as one manifestation of a broad principle of the evolutionary biology of risk-taking for a given potential benefit. They also suggest that disgust and related emotions can be usefully examined through the theoretical lens of decision making under risk in light of human evolution. 

The second aim of our study was to assess whether, in our sample, the emotion of disgust as measured by DS-R plays a role in future postgraduate studies orientation. Medical students who chose Internal Medicine as their future medical specialty scored higher in the core disgust subscale, indicating sensitivity to a real or perceived threat of oral incorporation and a reactive sense of offensiveness (spoiled milk, feces, and rats). Since Internal Medicine involves less exposure to bodily products compared to surgical and laboratory specialties, students with high core disgust are expected to prefer Internal Medicine to other specialties. 

Psychology students with Experimental postgraduate orientation scored lower in the animal reminder subscale, indicating lower sensitivity to attitudes and practices surrounding sex, injury of the body or violations of its outer envelope and death. This is in line with the fact that experimental psychology often includes high exposure to animal experiments, and reminders of death and of the fact that “we are animals”.

A logistic regression model for postgraduate Experimental orientation revealed that female psychology students had a lower probability of choosing the Experimental orientation. This might be explained by the fact that females are thought to be more empathetic and less systematizing compared to males [25], a trait that could make them choose a more clinically based orientation with more human contact. Psychology students with higher total disgust score were found to prefer clinical or educational to experimental orientation, probably because experiments that might include animals are more likely to expose them to disgust-eliciting stimuli. Finally, contrary to our expectations, the higher they scored in the core disgust subscale score, the higher the probability of choosing the Experimental orientation. It seems that some psychology students with high core disgust score prefer an orientation with less human contact and treatment. This is a finding that is difficult to interpret that has to be replicated. The analysis for medical students regarding the future option of Laboratory Specialty did not reveal any significant variable, indicating that in medical students, disgust propensity would not predict the choice of a specialty with less human contact and treatment. 

The present study has some limitations. First, there are limitations related to the comparability of the constructs of disgust across the two student groups. There is metric invariance, meaning that factor loads are equal across groups but only with partial strong (intecepts) and partial strict (residual) invariance, which suggests that respondents may attach different meanings to survey items. Previous researchers have noted, though, that the measurement invariance (MI) assumption is very hard to meet [26]. In particular, strict forms of MI referring to a situation in which measurement parameters are exactly the same across groups or measurement occasions, that is, an enforcement of zero tolerance with respect to deviations between groups rarely holds. Nevertheless, we cannot ignore that academic background might play a role in the way that participants interpret the construct of disgust, and one must be cautious when interpreting results deriving from the comparison of the two groups. We cannot exclude the possibility that differences are due to a different interpretation of the construct of disgust as measured by DS-R and not to a different attitude towards disgust. Other limitations of the study are related to the recruitment of participants, since it was a web-based survey and there may be a self-selection bias that potentially compromises the generalisability of the findings. In addition, as this process guaranteed the anonymity of participants, we cannot exclude the possibility that a student completed the questionnaire more than once. The modest number of participants is another limitation regarding the generalizability of results. Finally, we should consider that there are many other factors determining students’ decision to pursue undergraduate studies and postgraduate orientation for both psychology and medical students [14,15,16,17,27].

## 5. Conclusions

In conclusion, despite the limitations, it might be that students’ academic orientation is related to disgust propensity, with psychology students scoring higher than medical students in DS-R. Given that disgust propensity and sensitivity are significantly associated with the disease-avoidance functionality, a better understanding of disgust might be particularly helpful in shaping students’ knowledge about disease [3,28]. Moreover, as already mentioned by other researchers, further work is needed to elucidate the developmental pathways of disgust and to explore how predisposition interacts with social norms. Since disgust is plastic, being able to retune according to signals from within the body and from the social and biological environment [28], a follow-up study on the emotion of disgust among medical and psychology students during their pre-graduate studies might help to further assess the role of disgust in choosing postgraduate studies. Finally, it seems that individual disgust sensitivities should also be taken into consideration when providing guidance and taking post-graduate career specialization decisions for both psychology and medical students. 

## Figures and Tables

**Table 1 diseases-08-00043-t001:** Demographic characteristics and disgust scale—revised DS-R scoring in the total number of participants and according to faculty.

Variable	Total	Medical Students	Psychology Students	*p*-Value
	(*n* = 191)	(*n* = 94)	(*n* = 97)	
Age (years), mean, (SD)	20.15 (2.21)	20.31 (2.27)	20.00 (2.15)	0.336
Sex, *n* (%)				<0.001
Males	43 (22.51)	33 (35.11)	10 (10.31)	
Females	148 (77.49)	61 (64.89)	87 (89.69)	
Study Year, *n* (%)				0.023
1–2 years	121 (63.35)	52 (55.32)	69 (71.13)	
3–6 years	70 (36.65)	42 (44.68)	28 (28.87)	
Study Year, *n* (%)				<0.001
1	29 (15.18)	28 (29.79)	1 (1.03)	
2	92 (48.17)	24 (25.53)	68 (70.10)	
3	32 (16.75)	12 (12.77)	20 (20.62)	
4	8 (4.19)	4 (4.26)	4 (4.12)	
5	7 (3.66)	6 (6.38)	1 (1.03)	
6	23 (12.04)	20 (21.28)	3 (3.09)	
Core disgust, mean, (SD)	31.34 (7.53)	28.44 (7.08)	34.14 (6.89)	<0.001
Animal reminder, mean, (SD)	16.25 (6.94)	12.95 (5.50)	19.45 (6.70)	<0.001
Contamination disgust, mean, (SD)	8.66 (3.50)	8.19 (3.46)	9.11 (3.50)	0.069
Total score, mean, (SD)	56.25 (15.50)	49.57 (13.54)	62.71 (14.57)	<0.001

**Table 2 diseases-08-00043-t002:** Score distribution in relation to specialty orientation of medical students.

Variable	Internal Medicine	Surgery	Laboratory	*p*-Value
Core disgust, mean, (SD)	30.73 (5.30)	26.44 (7.80)	28.22 (7.87)	0.019
Animal reminder, mean, (SD)	14.28 (4.60)	11.76 (5.50)	13.00 (8.11)	0.108
Contamination disgust, mean, (SD)	8.28 (3.18)	8.16 (3.68)	8.00 (3.97)	0.973
Total score, mean, (SD)	53.27 (9.55)	46.36 (14.94)	49.22 (18.26)	0.061

**Table 3 diseases-08-00043-t003:** Score distribution in relation to postgraduate orientation of psychology students.

Variable	Clinical	Educational	Experimental	*p*-Value
Core disgust, mean, (SD)	34.80 (6.88)	34.05 (6.58)	30.45 (6.98)	0.154
Animal reminder, mean, (SD)	19.98 (6.33)	21.71 (5.62)	12.00 (6.13)	<0.001
Contamination disgust, mean, (SD)	9.40 (3.43)	9.52 (3.36)	6.64 (3.50)	0.043
Total score, mean, (SD)	64.18 (14.43)	65.29 (13.07)	49.09 (11.41)	0.003

**Table 4 diseases-08-00043-t004:** Score distribution according to sex.

Variable	Males	Females	*p*-Value
All students (*n* = 191)			
Core disgust, mean, (SD)	27.70 (7.00)	32.39 (7.37)	<0.001
Animal reminder, mean, (SD)	13.74 (6.56)	16.98 (6.90)	0.007
Contamination disgust, mean, (SD)	8.63 (3.32)	8.67 (3.57)	0.946
Total score, mean, (SD)	50.07 (14.73)	58.04 (15.31)	0.003
Medical Students (*n* = 94)			
Core disgust, mean, (SD)	26.67 (6.86)	29.39 (7.06)	0.074
Animal reminder, mean, (SD)	12.21 (6.39)	13.34 (4.97)	0.344
Contamination disgust, mean, (SD)	8.58 (3.15)	7.98 (3.63)	0.432
Total score, mean, (SD)	47.45 (14.40)	50.72 (13.03)	0.267
Psychology Students (*n* = 97)			
Core disgust, mean, (SD)	31.10 (6.66)	34.49 (6.87)	0.141
Animal reminder, mean, (SD)	18.80 (4.29)	19.53 (6.94)	0.746
Contamination disgust, mean, (SD)	8.80 (4.02)	9.15 (3.46)	0.767
Total score, mean, (SD)	58.70 (12.92)	63.17 (14.74)	0.361

**Table 5 diseases-08-00043-t005:** Odd ratios (OR) and corresponding 95% confidence intervals (95% CI) of univariate logistic regression for the selection of experimental psychology (*n* = 97 psychology students).

	Multivariable Analysis	
	Odds Ratio (95% C.I.)	*p*-Value
Age (per year)	1.09 (0.69–1.71)	0.712
Sex		0.023
*Males*	1.00	
*Females*	0.09 (0.01–0.1)	
Study Year		
*1–2*		
*3+*		
Core disgust	1.50 (1.13–2.00)	0.005
Animal reminder		
Contamination disgust		
Total score	0.76 (0.64–0.89)	0.001

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
