# Peer review of "The Emotion of Disgust among Medical and Psychology Students"

_diseases, 2020, doi:10.3390/diseases8040043_

Round 1

Reviewer 1 Report

Unfortunately the authors still refuse to address the point I raised.  If they do not report what the coefficient alpha was for their own data, then the readers have no idea if the scale was consistent for this sample.  Reporting past results are great, but the present sample should show the same results, so report them.

In addition, stating that you need to cite a scale in the abstract means providing the source also (give a citation).

Author Response

“Unfortunately the authors still refuse to address the point I raised.  If they do not report what the coefficient alpha was for their own data, then the readers have no idea if the scale was consistent for this sample.  Reporting past results are great, but the present sample should show the same results, so report them.

In addition, stating that you need to cite a scale in the abstract means providing the source also (give a citation).”

-Cronbach’s α is now reported for our own data.

In the method section: “We analysed the internal consistency of the items using Cronbach’s α after reversing the score of questions 1, 6 and 10 for which higher scores indicated less disgust.” (lines 127-128)

In the results section: “Cronbach’s α for the 27 items of the scale was 0.86” (line 142)

-The citation of the scale is provided in the abstract “Disgust Scale – Revised (DS-R) (Olatunji et al, 2008)” (line 27)

Reviewer 2 Report

The authors have addressed the issues raised in my previous review. 

Some minor concerns remain related to exposition. Specifically, the paper needs to be carefully edited, as there are quite a few grammatically incorrect and problematic sentence structures throughout. For example, 

“…DR-R has been examined in A sample …..” (page 3, line 117).

“The EFA method concluded to a 3-factor solution…” should be “The EFA method determined a 3-factor solution…. (page 3, line 118). 

There are many such instances throughout the entire paper.

Author Response

“The authors have addressed the issues raised in my previous review.  

Some minor concerns remain related to exposition. Specifically, the paper needs to be carefully edited, as there are quite a few grammatically incorrect and problematic sentence structures throughout.”

The paper has been edited. We rephrased some sentences in the abstract and made changes in the rest of the manuscript (lines 117, 118, 131, 163, 164)

Round 2

Reviewer 1 Report

Thank you for finally inserting the coefficient alpha values.  Not sure why that was so difficult.  It is standard to report if the scale being used is internally consistent with the sample being analyzed.  It is as standard as reporting a mean and a standard deviation.

Paper will still need some close reading/editing for English

Author Response

"Paper will still need some close reading/editing for Englishr"

Thank you for reniewing our manuscript.

Following your suggestion we performed a close editing of our manuscript for English and made minor changes in the folowing lines: 32,53,62,102,109,110,118,166,176,180,200-201,205,227,236,239,245,251

This manuscript is a resubmission of an earlier submission. The following is a list of the peer review reports and author responses from that submission.

Round 1

Reviewer 1 Report

With respect, I read the manuscript about The emotion of disgust among Medical and Psychology students, here are some suggestions for your consideration.

1) The abstract is rather vague to the topic of interest.
2) Introduction failed to provide sufficient scientific logic about he emotion of disgust and the target population. The emotion of disgust is not sufficiently explained. The logic is kind of messy, descriptive only and not deducive or convincing. 
3) Materials and Methods part did not provide enough information to understand how the study was conducted. There was no basic information about how and why to choose a a web-based survey, even though it is a convenience sample.All these make the audience hard to judge the quality of study. 
4) Some important results didn't report or analysis in detail (why the chosen approach, size effect, among others).
5) Discussion did not seem to be well related to the study results.  Overall, the contents of background, results, conclusions and discussion of this article are not very relevant to each other.

PS. I wish I could be more positive, but the manuscripts needs much improvement before a new submission. Morever, I do not understand why the authors have selected this journal and its readership.

Author Response

1) The abstract is rather vague to the topic of interest.

-We added a phrase and rephrased another sentence in order to make our ideas more explicit.

“The emotion of disgust evolved as a way to protect one’s self from illness and is associated to various aspects of disease avoidance”.

“Contextual factors, such as academic background, might have an impact on DS-R scoring, especially among medical students where the notion of disease plays a central role”.

-We also made minor changes in order to keep the number of words up to 200.

2) Introduction failed to provide sufficient scientific logic about he emotion of disgust and the target population. The emotion of disgust is not sufficiently explained. The logic is kind of messy, descriptive only and not deducive or convincing. 

-We added two phrases in the introduction section in order to provide the logic about the emotion of disgust and the target population.

“Students of a different academic background and especially medical students, who are expected to be exposed to diseases more than any other students, might score differently in DS-R.”

“Our hypotheses were that educational orientation, while implying a different attitude towards disease avoidance, would elicit different disgust scores …”

-We think that the emotion of disgust is sufficiently explained in the first 21 lines of the introduction.

3) Materials and Methods part did not provide enough information to understand how the study was conducted. There was no basic information about how and why to choose a a web-based survey, even though it is a convenience sample.All these make the audience hard to judge the quality of study. 

In the materials and method section (design and participants) we provided further information on how the study was conducted and the advantages of choosing a web-based survey.

“Participants were reached by one of their classmates through their University Social Media groups. Students are familiar with web-based approaches while having their classmates as contact persons instead of their professors possibly increases objectivity of the responses.”

4) Some important results didn't report or analysis in detail (why the chosen approach, size effect, among others).

In the Statistics and Results section there have been major changes according to all reviewers’ suggestions. We repeated the analysis and changes and the new results -similar to those reported in the original submission- are shown in the relevant sections and tables 1-4.

5) Discussion did not seem to be well related to the study results.  Overall, the contents of background, results, conclusions and discussion of this article are not very relevant to each other.

PS:… I do not understand why the authors have selected this journal and its readership.

By adding and changing some phrases through the whole manuscript we tried to clarify the logic of the study and increase the relevance of the various sections. One of the main ideas is that the emotion of disgust is related to disease avoidance and it would be interesting to explore it among students of disciplines (such as medicine) where the notion of disease is central. We also think that this idea makes the study relevant to the journal scopes.

In the discussion section we also added: “It might be that students of other academic backgrounds with less human contact and minimal exposure to physical or mental diseases would score differently-possibly even higher- in DS-R, reflecting a different attitude towards disease avoidance.”

In the conclusions section in order to make the text more concise and relevant to the rest of the study we omitted some sentences that seemed to be redundant

1) The abstract is rather vague to the topic of interest.

-We added a phrase and rephrased another sentence in order to make our ideas more explicit.

“The emotion of disgust evolved as a way to protect one’s self from illness and is associated to various aspects of disease avoidance”.

“Contextual factors, such as academic background, might have an impact on DS-R scoring, especially among medical students where the notion of disease plays a central role”.

-We also made minor changes in order to keep the number of words up to 200.

2) Introduction failed to provide sufficient scientific logic about he emotion of disgust and the target population. The emotion of disgust is not sufficiently explained. The logic is kind of messy, descriptive only and not deducive or convincing. 

-We added two phrases in the introduction section in order to provide the logic about the emotion of disgust and the target population.

“Students of a different academic background and especially medical students, who are expected to be exposed to diseases more than any other students, might score differently in DS-R.”

“Our hypotheses were that educational orientation, while implying a different attitude towards disease avoidance, would elicit different disgust scores …”

-We think that the emotion of disgust is sufficiently explained in the first 21 lines of the introduction.

3) Materials and Methods part did not provide enough information to understand how the study was conducted. There was no basic information about how and why to choose a a web-based survey, even though it is a convenience sample.All these make the audience hard to judge the quality of study. 

In the materials and method section (design and participants) we provided further information on how the study was conducted and the advantages of choosing a web-based survey.

“Participants were reached by one of their classmates through their University Social Media groups. Students are familiar with web-based approaches while having their classmates as contact persons instead of their professors possibly increases objectivity of the responses.”

4) Some important results didn't report or analysis in detail (why the chosen approach, size effect, among others).

In the Statistics and Results section there have been major changes according to all reviewers’ suggestions. We repeated the analysis and changes and the new results -similar to those reported in the original submission- are shown in the relevant sections and tables 1-4.

5) Discussion did not seem to be well related to the study results.  Overall, the contents of background, results, conclusions and discussion of this article are not very relevant to each other.

PS:… I do not understand why the authors have selected this journal and its readership.

By adding and changing some phrases through the whole manuscript we tried to clarify the logic of the study and increase the relevance of the various sections. One of the main ideas is that the emotion of disgust is related to disease avoidance and it would be interesting to explore it among students of disciplines (such as medicine) where the notion of disease is central. We also think that this idea makes the study relevant to the journal scopes.

In the discussion section we also added: “It might be that students of other academic backgrounds with less human contact and minimal exposure to physical or mental diseases would score differently-possibly even higher- in DS-R, reflecting a different attitude towards disease avoidance.”

In the conclusions section in order to make the text more concise and relevant to the rest of the study we omitted some sentences that seemed to be redundant

1) The abstract is rather vague to the topic of interest.

-We added a phrase and rephrased another sentence in order to make our ideas more explicit.

“The emotion of disgust evolved as a way to protect one’s self from illness and is associated to various aspects of disease avoidance”.

“Contextual factors, such as academic background, might have an impact on DS-R scoring, especially among medical students where the notion of disease plays a central role”.

-We also made minor changes in order to keep the number of words up to 200.

2) Introduction failed to provide sufficient scientific logic about he emotion of disgust and the target population. The emotion of disgust is not sufficiently explained. The logic is kind of messy, descriptive only and not deducive or convincing. 

-We added two phrases in the introduction section in order to provide the logic about the emotion of disgust and the target population.

“Students of a different academic background and especially medical students, who are expected to be exposed to diseases more than any other students, might score differently in DS-R.”

“Our hypotheses were that educational orientation, while implying a different attitude towards disease avoidance, would elicit different disgust scores …”

-We think that the emotion of disgust is sufficiently explained in the first 21 lines of the introduction.

3) Materials and Methods part did not provide enough information to understand how the study was conducted. There was no basic information about how and why to choose a a web-based survey, even though it is a convenience sample.All these make the audience hard to judge the quality of study. 

In the materials and method section (design and participants) we provided further information on how the study was conducted and the advantages of choosing a web-based survey.

“Participants were reached by one of their classmates through their University Social Media groups. Students are familiar with web-based approaches while having their classmates as contact persons instead of their professors possibly increases objectivity of the responses.”

4) Some important results didn't report or analysis in detail (why the chosen approach, size effect, among others).

In the Statistics and Results section there have been major changes according to all reviewers’ suggestions. We repeated the analysis and changes and the new results -similar to those reported in the original submission- are shown in the relevant sections and tables 1-4.

5) Discussion did not seem to be well related to the study results.  Overall, the contents of background, results, conclusions and discussion of this article are not very relevant to each other.

PS:… I do not understand why the authors have selected this journal and its readership.

By adding and changing some phrases through the whole manuscript we tried to clarify the logic of the study and increase the relevance of the various sections. One of the main ideas is that the emotion of disgust is related to disease avoidance and it would be interesting to explore it among students of disciplines (such as medicine) where the notion of disease is central. We also think that this idea makes the study relevant to the journal scopes.

In the discussion section we also added: “It might be that students of other academic backgrounds with less human contact and minimal exposure to physical or mental diseases would score differently-possibly even higher- in DS-R, reflecting a different attitude towards disease avoidance.”

In the conclusions section in order to make the text more concise and relevant to the rest of the study we omitted some sentences that seemed to be redundant "..., indicating a higher level of disease avoidance.  Since DS-R scale was constructed and refined mainly with young and largely female psychology student population the educational orientation should be taken into account when assessing disgust propensity in particular populations. "

Reviewer 2 Report

This paper looks at small samples of medical and psychology students who are mainly women in their responses to a disgust scale (the DS-R).  The analyses have been conducted incorrectly.  For some reason the authors have decided to compare medians and not means.  The authors fail to report the internal consistency value of the DS-R scales.  The reader doesn't even know if the scale is worth using or not.  Strangely the authors do not even compare their results to those previously reported using a Greek sample.  Means and independent-sample t-tests should have been used and not the strange middle 50%.  Analyses should be done properly and then presented.

Minor points:

  • "original of humans" should be "origins of humans"
  • careful with the "and" versus "&" ("and" is used within the text). 

Author Response

This paper looks at small samples of medical and psychology students who are mainly women in their responses to a disgust scale (the DS-R).  The analyses have been conducted incorrectly.  For some reason the authors have decided to compare medians and not means.  The authors fail to report the internal consistency value of the DS-R scales.  The reader doesn't even know if the scale is worth using or not.  Strangely the authors do not even compare their results to those previously reported using a Greek sample.  Means and independent-sample t-tests should have been used and not the strange middle 50%.  Analyses should be done properly and then presented.

Minor points:

  • "original of humans" should be "origins of humans"
  • careful with the "and" versus "&" ("and" is used within the text). 

-In the initial analysis we used median and 25th and 75th percentiles since they are measures of central tendency and dispersion, respectively, as the mean and standard deviation. We repeated the analysis providing mean and standard deviations and performed comparisons using the independent samples t-test. The results are similar to those reported in the original submission. They are presented in tables 1-4 and the appropriate corrections were made in the text as well.

-Psychometric properties of the Greek version of DS-R that we used in our study are now presented in the “Materials and Methods” section under the “DS-R scale”

“In the Greek version of DS-R scale, Cronbach’s alpha was 0.84 for the total scale reflecting satisfactory internal consistency. With respect to the test-retest reliability, the Interclass Correlation Coefficient (ICC) ranged between 0.85 and 0.92 (p< 0.001 in all cases), confirming the stability of the scale at each of the three factors.”

-The minor points in the introduction section have been corrected

This paper looks at small samples of medical and psychology students who are mainly women in their responses to a disgust scale (the DS-R).  The analyses have been conducted incorrectly.  For some reason the authors have decided to compare medians and not means.  The authors fail to report the internal consistency value of the DS-R scales.  The reader doesn't even know if the scale is worth using or not.  Strangely the authors do not even compare their results to those previously reported using a Greek sample.  Means and independent-sample t-tests should have been used and not the strange middle 50%.  Analyses should be done properly and then presented.

Minor points:

  • "original of humans" should be "origins of humans"
  • careful with the "and" versus "&" ("and" is used within the text). 

-In the initial analysis we used median and 25th and 75th percentiles since they are measures of central tendency and dispersion, respectively, as the mean and standard deviation. We repeated the analysis providing mean and standard deviations and performed comparisons using the independent samples t-test. The results are similar to those reported in the original submission. They are presented in tables 1-4 and the appropriate corrections were made in the text as well (means and SD instead of median and 25th and 75th percentiles).

-Psychometric properties of the Greek version of DS-R that we used in our study are now presented in the “Materials and Methods” section under the “DS-R scale”

“In the Greek version of DS-R scale, Cronbach’s alpha was 0.84 for the total scale reflecting satisfactory internal consistency. With respect to the test-retest reliability, the Interclass Correlation Coefficient (ICC) ranged between 0.85 and 0.92 (p< 0.001 in all cases), confirming the stability of the scale at each of the three factors.”

-The minor points in the introduction section have been corrected

Reviewer 3 Report

This study uses the Disgust Scale-Revised (DS-R) that is apparently characterized by three sub-scales (core, animal reminder, and contamination disgust) and examines the potential impact of the academic context on its scoring. The study therefore attempts to determine whether students of different academic backgrounds score differently on the DS-R. To do so, a sample of medical students is compared to a sample of psychology students and then reports on their observed differences.

Although determining whether differences between academic groups exist is clearly important, the assumption that the measurement instrument possesses the same construct validity across the groups is problematic. Before any legitimate comparisons of the actual scores can be made, it must first be established that the same constructs are being measured across the identified groups. In other words, the comparability of the constructs of disgust across the different student groups must first the established. This is extremely important because if the constructs measured by the DS-R are not similar across groups, there may be a lack of construct validity of the measurement scale for the particular groups in question and comparisons of their scores cannot be directly made.

To answer this question of score comparability, it is necessary to assess the measurement invariance of the measured scale across the two academic groups. Measurement invariance analyses across groups are generally conducted by using a sequence commonly endorsed in the literature (Millsap, 2011). Specifically, the following models must be tested across groups based on progressively more stringent invariance assumptions: (a) configural invariance (invariance of determined overall factor structure), (b) weak invariance (#a satisfied, plus invariance of factor loadings), (c) strong invariance (#b satisfied, plus invariance of item intercepts), (d) strict invariance (#c satisfied, plus invariance of item uniquenesses), (e) latent variance-covariance invariance (#d satisfied, plus invariance of latent variance-covariance), and (f) latent means invariance (#e satisfied, plus invariance of latent factor means). A lack of group differences in these tests would be supportive and lend legitimacy to the types of comparisons performed in this study.

References

Milsap, R.E. (2011). Statistical approaches to measurement invariance. London, UK: Routledge-Taylor & Francis.

Author Response

Although determining whether differences between academic groups exist is clearly important, the assumption that the measurement instrument possesses the same construct validity across the groups is problematic. Before any legitimate comparisons of the actual scores can be made, it must first be established that the same constructs are being measured across the identified groups. In other words, the comparability of the constructs of disgust across the different student groups must first the established. This is extremely important because if the constructs measured by the DS-R are not similar across groups, there may be a lack of construct validity of the measurement scale for the particular groups in question and comparisons of their scores cannot be directly made.

To answer this question of score comparability, it is necessary to assess the measurement invariance of the measured scale across the two academic groups. Measurement invariance analyses across groups are generally conducted by using a sequence commonly endorsed in the literature (Millsap, 2011). Specifically, the following models must be tested across groups based on progressively more stringent invariance assumptions: (a) configural invariance (invariance of determined overall factor structure), (b) weak invariance (#a satisfied, plus invariance of factor loadings), (c) strong invariance (#b satisfied, plus invariance of item intercepts), (d) strict invariance (#c satisfied, plus invariance of item uniquenesses), (e) latent variance-covariance invariance (#d satisfied, plus invariance of latent variance-covariance), and (f) latent means invariance (#e satisfied, plus invariance of latent factor means). A lack of group differences in these tests would be supportive and lend legitimacy to the types of comparisons performed in this study.

We have assessed the factor invariance for the three subscales of the disgust scale. The results are shown in the table below.

chi2      df         Ref. model        deltac hi2      delta df    P

Model 1                                    5.16      2          -

Model 2 metric (loadings)         6.67      4          1                      1.51                  2          0.470

Model 3 strong (intercepts)       29.79    6          2                      23.12                2            <0.001

Model 4 strict (residuals)           33.13    8          3                      3.34                  2          0.188

Model 5 plus factor means        62.88    9          4                      29.75                1            <0.001

Model 6 plus factor var             62.97    10        5                      0.09                  1          0.765

The chi-square difference between Model 1 and Model 0 is not statistically significant, thus indicating invariance of the factor loadings across the two groups. However, the chi-square difference between Model 2 and Model 1 is statistically significant which indicates that there is not complete invariance of the intercepts across the two groups.

We further assessed a model with partial strong invariance (intercepts). 

chi2      df         refmodel           deltachi2          deltadf P

Model 1                                    5.16      2          -

Model 2 metric (loadings)         6.67      4          1                      1.51                  2          0.470

Model 3 partial strong               9.18      5          2                      2.51                  1          0.113

Model 4 partial strict                 9.27      6          3                      0.09                  1          0.764

Model 5 plus factor means        39.0      7          4                      29.75                1            <0.001

Thus, we conclude that there is partial strict measurement invariance but not structural invariance

“We tested for measurement and structural invariance of the disgust scale based on the three subscales (Core disgust, Animal reminder and Contamination disgust) across the two groups of students participating to the study. We used a forward approach, starting with a model assuming total lack of invariance and adding restrictions for equality of specific parameters across groups (factor loadings, item intercepts, residual variances, factor means and factor variances). We used chi-square tests to compare the nested models. A paragraph was added in the methods section and another paragraph in the results section.

“We tested for measurement and structural invariance of the disgust scale based on the three subscales (Core disgust, Animal reminder and Contamination disgust) across the two groups of students participating to the study. We used a forward approach, starting with a model assuming total lack of invariance and adding restrictions for equality of specific parameters across groups (factor loadings, item intercepts, residual variances, factor means and factor variances). We used chi-square tests to compare the nested models.”

“While testing for measurement and structural invariance of the DS-R we identified metric invariance (equivalent factor loading), partial strong (intercepts) and partial strict (residuals) invariance of the disgust scale based on the three subscales.”

Round 2

Reviewer 1 Report

Authors have addressed my concerns

Reviewer 2 Report

Thank you for revising the manuscript.  I believe you misunderstood my comments about the coefficient alpha.  On page 3, section 2.3, you cite an alpha of 0.84 and a test-retest reliability value.  As you did not test the test-retest values, I am assuming that the 0.84 is from a different study?  If so, cite that fact, but also include what the coefficient alpha was for your data.  It is unclear if the scale "worked" for your sample.

In addition, I believe you should cite the scale in the Abstract.  Sorry that I missed that point in the first round.

Reviewer 3 Report

The revised version of the manuscript now includes some information concerning the assessment of  the measurement invariance of the measured scale across the two academic groups. The authors appear to have followed the appropriate procedures to test for measurement invariance and subsequently concluded that there were some aspects of partial measurement invariance. 

It is commonly acknowledged in the literature that in order to be able to contrast comparable observed means, we must have full scalar equivalence. So what are the practical implications of having determining so aspects of partial invariance on the three measured scales? Do the identified non-invariant measures complicate in any manner the interpretation of observed differences among the studied groups? More details need to be provided so that the reader can better assess the implications of the findings reported in this study.

This is an important issue that needs some additional clarity because if the underlying factors are fundamentally different, then there is no basis for interpreting the observed differences (it is similar to the “apples and oranges” problem).